# Anti-Cancer Activities of Nano Amorphous Calcium Phosphates toward Premalignant and Oral Cancer Cells

**DOI:** 10.3390/biomedicines12071499

**Published:** 2024-07-05

**Authors:** Evelina Herendija, Milica Jakšić Karišik, Jelena Milašin, Miloš Lazarević, Nenad Ignjatović

**Affiliations:** 1Multidisciplinary PhD Studies, University of Belgrade, Studentski Trg 1, 11000 Belgrade, Serbia; nherendija@yahoo.com; 2Implant-Research Center, School of Dental Medicine, University of Belgrade, Dr Subotica 8, 11000 Belgrade, Serbia; milica.jaksic@stomf.bg.ac.rs (M.J.K.); jelena.milasin@stomf.bg.ac.rs (J.M.); 3Institute of Technical Sciences of the Serbian Academy of Science and Arts, Knez Mihailova 35/IV, 11000 Belgrade, Serbia

**Keywords:** amorphous calcium phosphate, anti-cancer effect, oral cancer, dysplasia

## Abstract

Despite advancements in treatment, the squamous cell carcinoma (OSCC) patient survival rate remains stagnant. Conventional therapies have limited effectiveness, necessitating novel agents. Our study aims to synthesize and characterize amorphous calcium phosphate nanoparticles (nACPs), assess their potential cytotoxic effects on premalignant and malignant OSCC cells, and investigate possible mechanisms of action. The morphological features of nACP were investigated by field emission scanning coupled with energy dispersive spectroscopy (EDS), Fourier transform infrared spectroscopy (FTIR), and particle size distribution (PSD). Then, we examined the effect of nACPs on nanoparticle uptake, cell adhesion, viability, invasion ability, cell cycle, and gene expression. nACP uptake was dose-dependent, induced limited selectivity in cytotoxicity between healthy and malignant cells, and affected cellular adhesion and invasion. Early apoptosis was the predominant type of cell death. The nACP effect on viability was verified by alterations in the genes associated with apoptosis and proliferation. A high concentration of nACP was shown to arrest the cell cycle progression in the G0/G1 phase of both malignant and premalignant cells. This type of nACP justifies the development of a strategy for its potential use as an anti-cancer agent and/or anti-cancer active carrier for various drugs in oral cancer treatments.

## 1. Introduction

Cancer is a worldwide disease with a significant death rate. It is marked by anomalies in cellular function that result in the development, rapid growth, and survival of malignant cells [1]. Cancer is also associated with inflammation, immune system disorders, and the inhibition of apoptosis. Oral cancer, classified as a malignant neoplasm, impacts the lips, oral cavity, and oropharynx. Oral cavity cancer (OCC) is a type of head and neck cancer (HNC), and it ranks as the sixth most prevalent cancer globally, with over 300,000 new cases detected annually [2]. Pathological examination identifies the majority of OCCs as oral squamous cell carcinoma (OSCC), originating from squamous cells. It accounts for around 90% of oral cancers. Continual exposure to risk factors could result in the formation of oral potentially malignant disorders (OPMDs)—oral leukoplakia, erythroplakia, lichen planus, oral submucous fibrosis, actinic keratosis, and discoid lupus erythematosus are all common oral conditions that can develop into cancerous forms [3,4]. Prolonged exposure to risk factors facilitates the development of cancer by inducing genetic mutations, modified epigenetic alterations, and an imbalanced tumor microenvironment [4]. The five-year survival rate for patients with OSCC ranges from 40% to 50% [5]. Over the past three decades, the survival rate of individuals with OSCC has remained stagnant, despite advancements in diagnostic procedures and treatment regimens. The occurrence and frequency of OSCC are still on the rise, especially among younger individuals [6]. Given its multifactorial nature and intricate underlying mechanisms, treating this condition generally necessitates using multiple therapeutic techniques and combination therapy. Traditional cancer treatments include surgical procedures, chemotherapy, radiation therapy, hormone therapy, and immunotherapy. Chemotherapy and radiotherapy have minor efficacy in treating OSCC, and these methods are linked to significant side effects and a high likelihood of cancer relapse [7]. Therefore, it is crucial to prioritize endeavors related to cancer prevention, detection, and treatment.

Calcium phosphates (CaPs) are essential constituents of mammalian hard tissues, including bones and teeth. These materials, created in the laboratory to resemble human composition and structure, are widely used as bioceramic systems in reconstructive medicine. Because of their biocompatibility, bioactivity, and biodegradability, synthetic calcium-based biomaterials have been extensively studied for a diverse range of biological uses. Recently, the progress in CaP nanotechnology has led to a growing interest in nanostructured biomaterials in the biomedical domain, particularly in relation to cancer diagnosis, therapy, and theranostics [8].

Amorphous calcium phosphates Ca_3_(PO_4_)_2_×nH_2_O (ACPs) are a specific category of CaP materials that have been a considerable research challenge. The ACP phase is an intermediary phase in the manufacturing of various CaPs through precipitation. ACPs play a significant role in several biomaterials, such as coatings, cements, ceramics, and composites [9]. They can be readily transformed into poorly crystalline apatite, which is similar to the crystals found in bone minerals. Their strong reactivity can be utilized to create bioactive biomaterials. An important attribute of ACP is its significant propensity for spontaneous conversion into hydroxyapatite Ca_10_(PO_4_)_6_(OH)_2_ (HAp) [10]. ACP and HAp share the same chemical composition but differ in their structural groupings. A high-resolution examination of the crystal structure has shown that ACP is not entirely amorphous. Instead, it includes discernible pockets that are poorly crystalline and uniformly distributed within the amorphous matrix. This aligns with the metastable characteristic of ACP, which has a propensity to undergo spontaneous crystallization into HAp when conditions that promote this transition, such as high temperature and/or humidity, are present [11]. The qualities and possible applications of ACP and HAp phases can be significantly influenced by their morphology and particle size, in addition to their composition and structure. In the past ten years, advancements in nanotechnology have made it possible to create ACP and HAp particles that are small, measuring between 1 and 100 nanometers in size. Particles of these dimensions can greatly change their characteristics and, therefore, optimize their utilization in medicine and dentistry. Studies have confirmed that nanosized and nanocrystalline calcium orthophosphates can replicate the size of the individual parts seen in calcified tissues; therefore, they can be employed in bimetallization and as biomaterials because of their enhanced biocompatibility [12]. ACP nanoparticles (nACPs) play a crucial role in the colonization of microspheres by stimulating the synthesis of proteins that facilitate the adsorption of the nanoparticles to the tissues and initiate the interaction between the nanoparticles and the biological environment [13]. Inside the tumor microenvironment, the presence of nACP can induce apoptosis and reduce cell proliferation. The mechanism of this connection has been elucidated from several perspectives. The activation of apoptosis pathways depends on the adsorption and internalization of nanoparticles in cancer cells, leading to the activation of programmed cell death mechanisms [14]. Controlling the properties of ACP nanoparticles is essential for the intended application; however, the specific method by which HAp nanoparticles are internalized into cells has not yet been identified.

The main objective of this work was to create nanosized amorphous calcium phosphate particles that are both biocompatible and have anti-cancer effects on oral malignant and premalignant cells. This research has the potential to provide the groundwork for the advancement of techniques that use selective nACPs as carriers for different medications. This might potentially improve the efficacy of oral cancer treatment.

Therefore, the specific aims of this study were to design nACPs; to assess the potential selective cytotoxic effects of nACPs on healthy, premalignant, and malignant cell lines; and to investigate several possible mechanisms of nACP action—including the effect on cell adhesion, invasion, apoptotic and proliferation gene expression, the induction of apoptosis, and cell cycle arrest.

## 2. Materials and Methods

### 2.1. Synthesis of ACP Powder

A previously developed procedure [11] was adapted for the synthesis of ACP powder. ACP was made by abruptly adding a solution containing 23.6 g of Ca(NO_3_)_2_ × 4H_2_O (Sigma Aldrich, Berlin, Germany) in 200 mL of distilled water and 24 mL of 28% NH_4_OH into a solution comprising 6 g of NH_4_H_2_PO_4_ and 10 mL of 28% NH_4_OH in 200 mL of distilled water. The fine precipitate formed upon mixing had been aged for 15 s before it was collected. The sol was centrifuged and washed multiple times until pH 7 was attained. The resulting sol was then poured into Petri (glass) dishes. Finally, the obtained powder was subjected to two-step lyophilization, first at −10 °C and 0.37 mbar for 1 h, and then at −54 °C and 0.1 mbar for 4 h. After lyophilization, the obtained dry powder was transferred to plastic tubes (Flex-Tube^®^, Eppendorf, Hamburg, Germany).

#### Powder Characterization

The morphological features of the particles were investigated by field emission scanning (Carl Zeiss ULTRA Plus FE-SEM, Berlin, Germany) coupled with energy-dispersive spectroscopy (EDS). Fourier transform infrared spectroscopy (FTIR) was performed on a Nicolet iS10 FTIR Spectrometer with a Smart iTR Diamond Attenuated Total Reflectance accessory (Thermo Scientific Instruments, Waltham, MA, USA) in the spectral range from 400 to 4000 cm^−1^. The particle size distribution (PSD) was measured on 10 mg/mL of powders dispersed in water using a Mastersizer 2000 (Malvern Instruments Ltd., Malvern, UK) and a HydroS dispersion unit for liquid dispersants.

### 2.2. Biological Assays

#### 2.2.1. Cell Cultures

Two different cell types were used in the experiments: dysplastic oral keratinocytes (DOKs)—the premalignant cell line—and the SCC-25 oral cancer cell line.

Complete growth medium containing Dulbecco’s Modified Eagle Medium (DMEM) supplemented with 10% fetal bovine serum (FBS) and 1% antibiotic–antimitotic solution (all from Thermo Fisher Scientific, Waltham, MA, USA) was used to culture the SCC–25 (ATCC^®^, CRL–1628™, Manassas, VA, USA) cell line and the premalignant DOK cell line (European Collection of Authenticated Cell Cultures, 94122104). The cells were kept in a humidified environment with 5% CO_2_ at 37 °C. Every two to three days, a new culture growth medium was used. Cell cultures were passaged after reaching 80% confluence. Third-passage cells were used for the experiments. Hydrocortisone (0.5 µg/mL; Thermo Fisher Scientific, Waltham, MA, USA) was added to the cell culture medium for the DOK cell line, while 400 ng/mL was added to the cell culture medium for the SCC-25 cell line.

#### 2.2.2. Nanoparticle Uptake

Six-well plates containing 10^6^ cells/mL of the complete growth media were used to plate cells for the flow cytometry analysis of nanoparticle uptake. After 24 h without ACPs, cells were subjected to a 24 h treatment with varying concentrations of nACPs. Following three five-minute PBS washes, the cells were trypsinized while still in the proliferative growth phase, centrifuged, and then raised in 1 mL of cold PBS containing 10% FBS before being promptly examined. A BD FACSMelody^TM^ (BD Biosciences, San Jose, CA, USA) flow cytometer with a 488 nm laser, a forward scatter light (FSC) diode detector, a side scatter light (SSC) detector on a photomultiplier tube, and BD FACSChorus^TM^ software (Ver. 1.3.3) was used. These measurements are influenced by the flow rate; hence, modest flow rates were always used. The cytometer was configured to measure FSC linearly and SSC logarithmically. To determine the range for the maximum SSC signal and the minimum FSC signal [15], the highest concentration of ACP nanoparticles (4 mg/mL) was analyzed initially. There were three iterations of the experiment. 

#### 2.2.3. Cell Adhesion Assay

Cells (1 × 10^4^/mL) were seeded in culture plates containing varying nACP concentrations. As a control, the same number of cells was plated in culture plates without ACPs. Each group utilized a total of six wells, and the cells were grown in an incubator with 5% CO_2_ at 37 °C. After 24 h, non-adherent cells were counted using a hemocytometer.
Adhesion rate (%) = (number of seeded cells − non–adhered cells)/(number of seeded cells) × 100(1)
was the formula [16,17] used to calculate non-adherent cell numbers after the 24 h treatment. There were three iterations of the experiment.

#### 2.2.4. MTT Assay

A 96-well plate was seeded with 1 × 10^4^ cells per well. After 24 h, the cells were then given 100 µL of complete medium supplemented with 1, 2, and 4 mg/mL of ACPs and incubated for 1, 3, or 7 days. The culture medium containing ACPs was replaced every 3 days. After the treatment, the supernatant was removed and washed with 100 µL of PBS and 100 µL of the MTT (3-(4,5-dimethylthiazol-2-yl)- 2,5-diphenyltetrazolium bromide) solution (Sigma-Aldrich Inc., Burlington, MA, USA). The precipitates were shaken at 37 °C to dissolve them in 100 µL of dimethyl sulfoxide (DMSO) (Sigma-Aldrich Inc., MA, USA). Optical density (OD) was measured at 540 nm using a microplate reader (RT-2100c, Rayto, Shenzhen, China). The formula for calculating cell viability (%) was
(ODsample − ODblank)/(ODcontrol − ODblank) × 100 (2)

There were three iterations of the experiment.

#### 2.2.5. Apoptosis Assay (Annexin V)

Cells were seeded into 24-well plates with a density of 1 × 10^5^ per well and cultured with varying concentrations of nACPs. Following a 24 h incubation period, apoptosis detection was carried out using an Annexin V-FITC Apoptosis Detection Kit (Invitrogen, Thermo Fisher Scientific, Waltham, MA, USA) according to the manufacturer’s instructions. The Annexin V-FITC staining was analyzed using the BD FACSMelody^TM^ flow cytometer (BD Biosciences, San Jose, CA, USA), and the results were displayed in a two-dimensional dot plot of propidium iodide (PI) versus Annexin V-FITC. PI was utilized to identify viable, necrotic, late, or early apoptotic cells. The plots were categorized into four regions based on the characteristics of the cells. These regions included (a) viable cells, which were negative for both probes (PI/FITC −/−); (b) apoptotic cells, which were PI-negative and Annexin-positive (PI/FITC −/+); (c) late apoptotic cells, which were both PI- and Annexin-positive (PI/FITC +/+); and (d) necrotic cells, which were PI-positive and Annexin-negative (PI/FITC +/−). The cells cultured in the complete growth medium under identical conditions were utilized as controls. There were three iterations of the experiment.

#### 2.2.6. Cell Cycle Determination

The cells were prepared to ensure a minimum of 10^6^ cells per test sample in a 6-well plate. Subsequently, they were centrifuged at 1400 rpm for 6 min and washed with PBS for 5 min. Following this, centrifugation was performed again, after which, the cells were suspended in 300 µL of PBS. Gradually, 700 µL of 96% ice-cold ethanol was added drop-wise, followed by an incubation period of 2 h at 4 °C. Further centrifugation at 1700 rpm for 6 min was conducted, and ethanol was removed, with cells resuspended in PBS. After another centrifugation under the same conditions, the cells were resuspended in 500 µL of PBS, and 7 µL of RNase A (from a stock of 100 µg/mL) was added, followed by 15 min of incubation at 37 °C. Just before the analysis on the flow cytometer, PI was added at a final concentration of 50 µg/mL. Finally, the percentage of cells in different phases of the cell cycle was determined using the BD FACSMelody^TM^ flow cytometer and BD FACSChorus^TM^ software (Ver. 1.3.3). There were three iterations of the experiment.

#### 2.2.7. RNA Isolation and Reverse Transcription

RNA was extracted from a sample of cells (10^6^ per well) treated with the lowest concentration of nACPs (1 mg/mL) for 24 h in a 6-well plate using a TRIzol reagent (Invitrogen, Thermo Fisher Scientific, Waltham, MA, USA), following the manufacturer’s instructions. The RNA concentration was measured using a microvolume spectrophotometer from Shimadzu Scientific Instruments (Carlsbad, CA, USA). An oligo d(T) primer and the RevertAid First Strand cDNA Synthesis Kit (Thermo Fisher Scientific, Waltham, MA, USA) were used to successfully synthesize cDNA from 2 µg of total RNA [18].

#### 2.2.8. qPCR

A real-time quantitative polymerase chain reaction (qPCR) was conducted using the first-strand 2 µL cDNA, 0.75 µM of forward and reverse primers, and 7.5 µL of a SensiFAST SYBR Hi–ROX Kit (Bioline, London, UK). The expressions of the following markers were analyzed: CASP3, Cyclin D, BAX, BCL2, and β-Catenin. A reference gene, glyceraldehyde-3-phosphate dehydrogenase—GAPDH, was utilized for the study. Gene expression values were calculated using the 2^−∆Ct^ method [19]. All the primer sequences used in this study can be found in Table 1. There were three iterations of the experiment.

#### 2.2.9. Statistical Analysis

The analyses were conducted using GraphPad Prism version 9 software (GraphPad Software, Inc.). Following the Kolmogorov–Smirnov normality test analysis of the distribution, a single pooled variance ordinary one-way ANOVA and post hoc Tukey’s multiple comparisons test or independent-sample T-tests were conducted. The data are shown as the mean ± SD. * *p* < 0.05, ** *p* < 0.01, *** *p* < 0.001, and **** *p* < 0.0001 denote statistical significance.

## 3. Results

### 3.1. ACP Powder Characterization

The basic physicochemical characterization of the synthesized ACP powder is presented in Figure 1. The dry ACP powder existed as a partial agglomeration (Figure 1a) with primary particles sized <100 nm. EDS analysis of position 1(+) in Figure 1a confirmed the presence of constituent atoms of ACPs (Figure 1b). The aqueous suspensions of the particles of ACPs were analyzed to establish their size distributions. Figure 1c shows the size distributions for the suspended particles of ACP powders in distilled water. The particle sizes for the analyzed powder ranged from 22 nm to 1 μm. The particles had a relatively uniform distribution of sizes, with the d50 parameter equaling 68 nm. The powder contained a small portion of larger and smaller particles, d90 = 147 nm and d10 = 32 nm, respectively. An FTIR analysis of the ACP powder (Figure 1d) confirmed the qualitative composition of the ACPs with the characteristic assignments of groups (OH and PO_4_), consistent with our previous research. The validation of the amorphous phase of the ACPs was confirmed with XRD diffraction patterns, which was also shown in our previous research [11].

### 3.2. Cell Adhesion Ability after nACP Treatment

Figure 2 shows graphs quantifying cell adhesion, revealing a significant dose-dependent relationship with ACP nanoparticle concentration following a 24 h treatment period. The percentage of cells adherent to the plastic substrate notably decreased in both cell lines compared to the control (untreated cells), indirectly suggesting that nACP induces cell death.

The adhesion rate of untreated DOK was 87% ± 6.02, whereas cells treated with 1, 2, and 4 mg/mL nACP exhibited a decrease in the mean adhesion rate of 47% ± 6.12, 40% ± 7.22, and 20% ± 4.21, respectively (Figure 2a). SCC-25 cells exhibited a similar decreasing trend, with statistically significant differences between the control (88% ± 6.33) and 1, 2, and 4 mg/mL nACP-treated cells (80% ± 5.03, 73% ± 8.12, and 47% ± 6.22, respectively) (Figure 2b).

### 3.3. Cytotoxic Effect of ACP Nanoparticles

The cytotoxic effect of ACP nanoparticles on oral premalignant and cancer cells was evaluated 1 day, 3 days, and 7 days after the treatment. Generally, a reduction in cell viability was observed in a dose- and time-dependent manner. Both DOK and SCC-25 cells exhibited a significant decrease in viability. After the first day of the treatment, both cell lines treated with varying concentrations of ACP nanoparticles (1, 2, and 4 mg/mL) showed a decrease in viability for DOK (93% ± 6.46, 83% ± 2.84, 87% ± 1.13, respectively) and SCC-25 cells (98% ± 19.21, 71% ± 5.77, 32% ± 4.38, respectively); 72 h after the treatment, the viability of SCC-25 cells decreased significantly (75% ± 10.42, 50% ± 13.29, 26% ± 2.76, respectively). Further inhibition of cell viability was observed 7 days after the treatment in DOK (82% ± 10.71, 73% ± 5.17, 78% ± 4.70, respectively) and SCC-25 (76% ± 11.10, 46% ± 16.60, 16% ± 2.06, respectively) cell lines (Figure 3). The cytotoxic effect was the most evident in the SCC-25 cells, which were more sensitive to nACPs.

### 3.4. ACP Nanoparticle Uptake Analysis

The quantification of the ACP nanoparticle cellular uptake following a 24 h cell treatment period is illustrated in Figure 4. After the treatment, cells exhibited a dose-dependent response, indicating the propensity of nanoparticles to adhere to cell membranes (or enter the cellular cytoplasm) as the particle concentration rises. Figure 4a,c represent visual qualitative cytograms of the uptake of nACP concentrations of 1, 2, and 4 mg/mL by DOK and SCC-25 cells. Figure 4b,d show the quantification of interaction between nACP nanoparticles and DOK and SCC-25 cells, based on the events ratio linearly and 1, 2, and 4 mg/mL nACPs logarithmically. The cytograms and graphs demonstrate a dose-dependent rise in side scatter, which correlates with the complexity of a cell in both cell cultures.

### 3.5. ACP Nanoparticle Effect on Apoptosis

As shown in Figure 5a,c, the Annexin analysis plots were partitioned into four regions, delineating the percentages of live (bottom left), necrotic (up left), and apoptotic (early in the bottom right and late in the upper right quadrant) cells. An induction of cell death was observed in DOK and SCC-25 cells treated with ACP nanoparticles. The findings suggest that early apoptosis is the predominant type of cell death triggered by ACP nanoparticles. The proportion of cell apoptosis revealed a dose-dependent relationship. Specifically, at treatment concentrations of 1, 2, and 4 mg/mL, apoptotic (early and late) cells accounted for 7%, 18%, and 26% of DOK cells (Figure 5b). Similarly, the results for SCC-25 cells with the same treatments were similar: apoptotic (early and late) cells in 1, 2, and 4 mg/mL nACP treatments accounted for 11%, 21%, and 35%, respectively (Figure 5d).

### 3.6. The Effect of nACP on the Cell Cycle

The impact of ACP nanoparticles on the cellular cycle of DOK and SCC-25 cells was assessed using a cell cycle analysis. This analysis is essential for understanding the phases of the cell cycle (sub-G1, G0/G1, S, and G2M phases) and detecting subcellular-level alterations over the course of incubation. We observed a significant percentage change in cells across different cell cycle stages (Figure 6). In the case of DOKs (Figure 6a), both the untreated (control) and cells treated with 1 and 2 mg/mL nACP predominantly exhibited the G2 phase (86% ± 2.33—control, 89% ± 3.12—1 mg/mL, and 86% ± 3.33—2 mg/mL). However, DOKs treated with 4 mg/mL of ACP nanoparticles demonstrated a G0/G1 phase arrest (94% ± 7.02). Similarly, SCC-25 cells (Figure 6b) displayed a comparable distribution, with a predominant G2 phase observed in untreated and 1 mg/mL concentration-treated cells (75% ± 5.10 and 87% ± 6.22, respectively). However, SCC-25 cells treated with 2 mg/mL nACP exhibited a 63% ± 2.42 cell population in the G0/G1 phase, while after a treatment with 4 mg/mL, 92% ± 3.98 of cells were in this phase.

### 3.7. Gene Expression Analysis of Apoptotic, Proliferative, and Oncogenic Markers after the nACP Treatment

In order to understand the molecular mechanism by means of which nACP induces apoptosis in human precancerous DOK and cancerous SCC-25 cells, we analyzed the expression levels of CASP3, Cyclin D, BAX, and BCL-2. The PCR expression profile of these genes in the DOK cell culture demonstrated that there was a statistically significant increase in CASP3, Cyclin D, and BCL2 gene expressions. A similar trend was noted in the CASP3 gene expression of the SCC-25 cell line, but without reaching statistical significance. The proapoptotic BAX gene was not significantly affected by the nACP treatment in both premalignant and malignant cell lines. Interestingly, β-Catenin, a gene which has an important role in the Wnt/β-Catenin signaling pathway, demonstrated significant downregulation after the nACP treatment (Figure 7).

## 4. Discussion

The identification of materials with promising inhibitory effects on cancer cells and tolerable side effects in healthy cells is vital for improving the current therapeutical approach to oral cancer. Recent studies suggest the use of CaPs for OSCC and lung and breast cancer therapy through controlled drug release and targeted drug delivery agents in vitro [20,21,22,23,24,25]. However, very few studies have examined the effects of CaP nanoparticles alone in in vitro conditions [24]. To the best of our knowledge, no studies have assessed the effect of nACPs on premalignant and oral cancer cells. We have demonstrated that the incubation of DOK and SCC-25 cells with nano amorphous calcium phosphates smaller than 100 nm has limited anti-cancer effects.

Firstly, cell adhesion ability was used as a preliminary indicator of cell viability. The effect of ACP nanoparticles on cells depended on concentration. Indirectly, by counting non-adherent cells, it was demonstrated that the number of adherent cells decreased as the nACP concentration increased. This trend was noticed in both cell cultures.

The cytotoxicity assessment in this study shows that nACPs inhibited cell viability in a dose- and time-dependent manner. Previous studies are in line with our study in terms of the dose-dependent cytotoxicity of ACPs [26,27,28]. Possible mechanisms of cytotoxicity could be the simple agglomeration and subsequent sedimentation of nanoparticles at relatively high concentrations, which might result in mechanical damage to cells [28]. On the other hand, Liu et al. demonstrate that HAp nanoparticles degrade in the lysosomes of cells, which increases intracellular Ca^2+^, leading to lysosomal rupture and inducing cell death through necrosis [26]. Finally, the mechanism could be associated with the protein synthesis decrease or the lack of key proteins for cell cycle progression [27]. Building on the previous results, we hypothesize that the possible nACP mechanism of action could be (i) mechanical damage through intracellular sedimentation, (ii) inducing cell necrosis and/or apoptosis, or (iii) possible disturbance in the cell cycle.

Our results related to the nanoparticle uptake could explain cellular death through mechanical damage. Indeed, based on cytogram visualization and the resulting quantification, it has been demonstrated that both cell lines show comparable dose-dependent uptake manners, while SCC-25 cells show a higher uptake level, especially with the highest nACP concentration (4 mg/mL). In the study which assessed the effect of hydroxyapatite nanoparticles on osteoblasts, it was demonstrated that nanoparticles with different concentrations, charges and sizes have different amounts of particles internalized into cells [29].

Using the Annexin V apoptosis assay (Figure 5), the cell death of DOK and SCC-25 cells incubated in direct contact with three different nACP concentrations for 24 h was detected. A flow cytometry analysis has shown that the density plots in the bottom right quadrant increase (FITC-Annexin V positive) as nACP concentration rises, meaning that early apoptosis is the predominant cell death type in both cell lines. The adhesion and MTT assays, which measure the cell viability and the metabolic activity of the cells, correlate with these results. The SCC-25 cell line (Figure 5c,d) exhibited higher levels of cell death, primarily early apoptosis. In contrast, more than 70% of DOK cells remained viable. The early detection of apoptosis suggests that the biomaterial may cause cellular stress, even if the metabolic function appears unaffected in the initial phases [30].

After the treatment with different concentrations of ACP nanoparticles for 24 h, the accumulation of DOK and SCC-25 cells in different cell cycle phases was observed (Figure 6). Our data suggest that the inhibition of cell growth by ACP nanoparticles mainly elicits a change in the redistribution of DOK and SCC-25 cells along the dose-dependent cell cycle arrest at the G0/G1 phase. As determined by the cell cycle analysis, the increased apoptosis rate is linked to the cell cycle arrest. The cell cycle is a complex process in which cells receive different growth-controlling signals integrated and processed at various points known as checkpoints [31]. G1 is the stage where the cell prepares to divide and proceed into the S phase, where cell DNA synthesis occurs. The G1 phase occurs after mitosis, which is a period during which the cell is responsive to both positive and negative growth signals. G2 is the period following the S phase in which the cell undergoes preparations for entering mitosis [32]. The primary mechanism of the anti-cancer impact of nACPs could be a significant increase in endocytosis within cancer cells, along with the suppression of protein synthesis [27]. The excessive amount of nACP taken up by cancer cells in the endoplasmic reticulum can inhibit protein synthesis by reducing the attachment of mRNA to the ribosome—this can also halt the cell cycle in the G0/G1 phase [27].

The activation of apoptosis pathways depends on the adsorption and internalization of nanoparticles in cancer cells [14]. Chemotherapy and radiation can trigger cellular death by activating the intrinsic, mitochondria-mediated mechanism of apoptosis. The activation of this route leads to the activation of the caspase protease cascade, which ultimately causes the internal death of the cell [33]. The progression of OSCC depends on two pathways, proliferation and apoptosis, which intersect at various points. There is a significant overlap in signaling between these two networks, meaning that any alterations in one network will impact the other [34]. According to our findings, one mechanism of action of the nACP anti-cancer effect is to disrupt the normal process of apoptosis. Apoptosis often follows a series of sequential processes that trigger or inhibit specific genes and proteins. It culminates in the activation of proteases, the fragmentation of DNA, and cell death [35]. The development of cancer mostly relies on the misbalance between pro-apoptotic proteins (e.g., BAX) and anti-apoptotic proteins (e.g., BCL2) [36]. On the other hand, the gene related to the proteolytic enzyme Caspase 3 (CASP3), which triggers DNA degradation, leading to cell death, also has great importance. The qPCR results (Figure 7) of our study reveal significant increases in the mRNA levels of the apoptotic marker CSP3 and BCL2 expressions in both premalignant and cancer cells. This study observed the induction of apoptosis following the nACP treatment due to the deregulation of gene expression. The initiation of apoptotic pathways may be facilitated by mitochondria through the upregulation of CASP3 expression [37]. Apoptosis encompasses both the intrinsic and extrinsic mechanisms—mitochondria govern the intrinsic pathway, while death receptors govern the extrinsic pathway [38]. Cyclin D is often overexpressed in a variety of cancers and is associated with tumorigenesis and metastasis. Cyclin D plays a crucial role in regulating the transition from the G1 to the S phase, and its upregulation has been associated with various types of cancer, such as breast, esophageal, bladder, and lung malignancies [39]. During the initial phases of OSCC, there is an increase in the expression of this gene, which leads to OSCC cell proliferation. In addition, Cyclin D is associated with decreased overall survival and unfavorable prognosis in individuals with OSCC [4]. Cyclin D expression on oral cancer cells could be associated with our cell cycle analysis results, where the cell cycle G1 phase arrest contributed to the regulation of Cyclin D expression. β-Catenin is an oncogenic protein promoting cell proliferation, migration, and invasion through the Wnt/β-Catenin signaling pathway. This pathway plays a crucial role in determining OSCC cell proliferation [4]. Our findings indicate that nACP caused a substantial decrease in the activity of the Wnt/β-Catenin signaling pathway in both cell lines, which contributed to our result related to cytotoxicity.

The main limitation of our study is the fact that it primarily investigates the effects of nACPs on premalignant and cancer cells. It could be beneficial to explore potential interactions with other components of the tumor microenvironment (i.e., immune cells, stromal cells, and the extracellular matrix). Understanding how nACPs influence the tumor microenvironment could provide additional insights into their therapeutic potential, and this topic will be addressed in the future research of our group. Additionally, there is a lack of in vivo validation. Translating our findings to in vivo models and eventually to clinical settings would be necessary to confirm the efficacy of this treatment approach. In vivo studies would provide a more comprehensive understanding of how nanoparticles interact with the complex biological environment, including systemic circulation, immune responses, and the tumor microenvironment.

## 5. Conclusions

In this study, the designed ACP nanosized particles induced limited cytotoxicity against premalignant and malignant OSCC cells and affected their adhesion. The cellular uptake of nanoparticles was observed, and early apoptosis was the predominant type of cell death triggered by nACPs. Higher concentrations of nACPs led to a halt in cell cycle progression in the G0/G1 phase of both malignant and premalignant cells. The effect of nACPs on proliferation, apoptosis, and cell cycle was further elucidated by the upregulation of apoptotic and cell cycle-related genes CASP3, BCL2, and Cyclin D and the downregulation of β-Catenin.

These results suggest that nACPs can inhibit the proliferation of OSCC cells with different mechanisms and have potential applications in cancer treatment. The collected evidence verifies that nACPs can induce apoptosis in various cancer cell lines and that they may represent a promising anti-cancer drug strategy for OSCC treatment.

In conclusion, the results indicate that this type of nACP justifies developing a strategy for its potential use as an anti-cancer agent and/or as a possible anti-cancer active carrier for various drugs in oral cancer treatments.

## Figures and Tables

**Figure 1 biomedicines-12-01499-f001:**
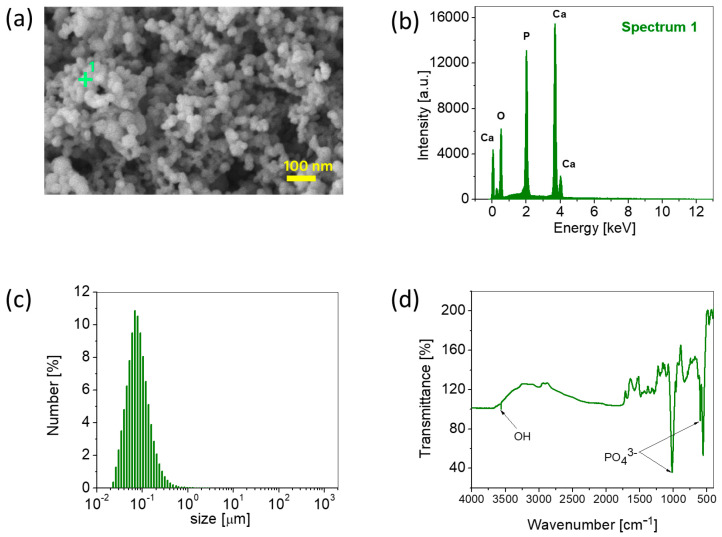
ACP powder characterization. (**a**) FESEM of ACP, (**b**) EDS of ACP (spectrum no 1), (**c**) PSD of ACP powder, (**d**) FTIR of ACP powder.

**Figure 2 biomedicines-12-01499-f002:**
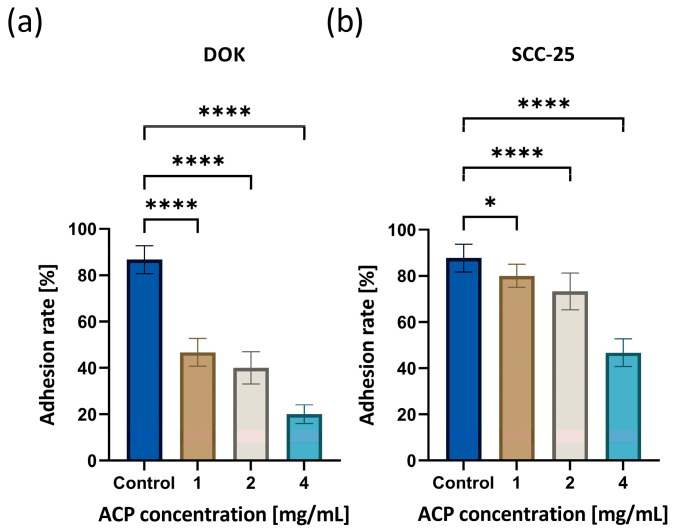
Cell adhesion assay of (**a**) DOK and (**b**) SCC-25 cells during 24 h. Control–untreated cells. * *p* < 0.05, and **** *p* < 0.0001. Data bars denote mean (*n* = 18) and error bars the standard deviation.

**Figure 3 biomedicines-12-01499-f003:**
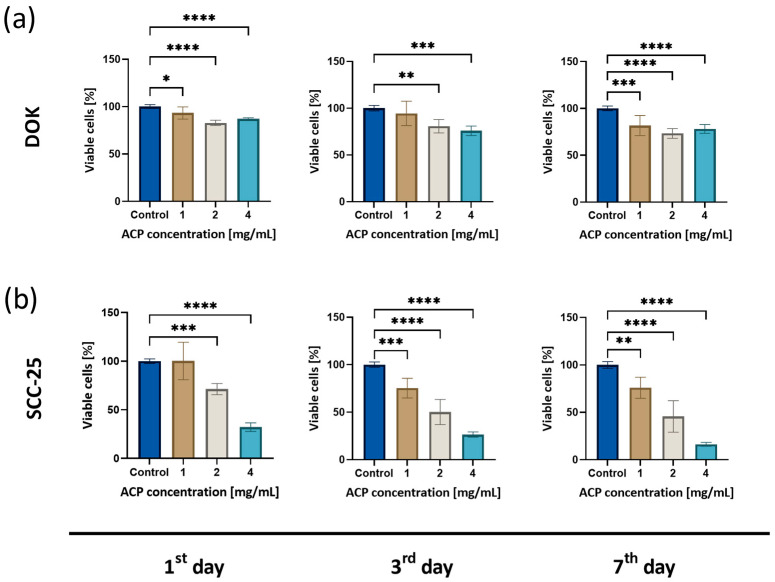
MTT assay of (**a**) DOK (**b**) and SCC-25 cells after the treatment with nACP for 1, 3, and 7 days. * *p* < 0.05, ** *p* < 0.01, *** *p* < 0.001, and **** *p* < 0.0001. Data bars denote mean (*n* = 18), and error bars, the standard deviation.

**Figure 4 biomedicines-12-01499-f004:**
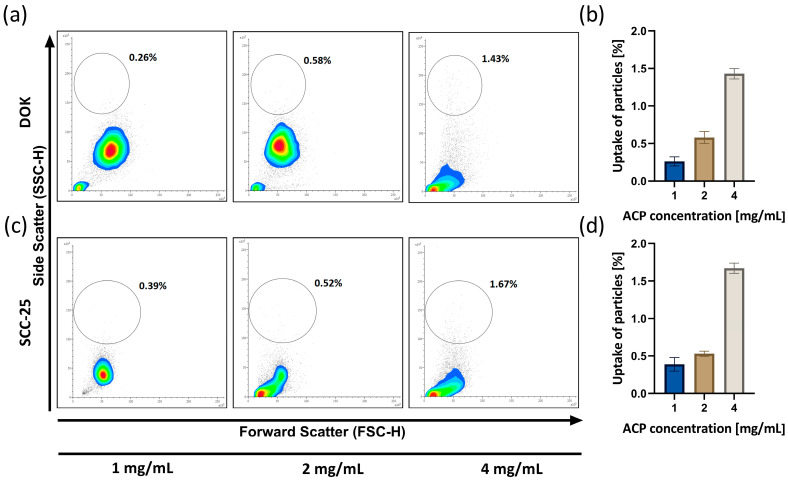
Cytograms (**a**) and quantification (**b**) of nACP uptake by DOK and cytograms (**c**) and quantification (**d**) of nACP uptake by SCC-25 cells after 24 h nACP treatment. Cytograms were shown as density plots, with the red color representing the highest cell density. Data bars denote mean (*n* = 3), and error bars, the standard deviation.

**Figure 5 biomedicines-12-01499-f005:**
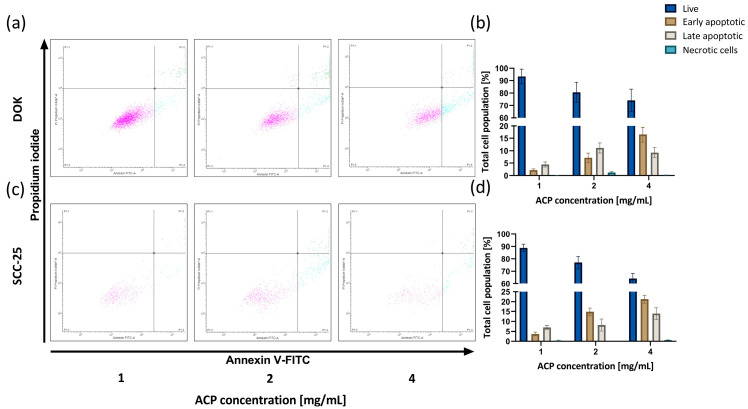
Annexin V flow cytometry assay evaluated apoptosis/necrosis of (**a**,**b**) DOK and (**c**,**d**) SCC-25 cells induced by 1, 2, and 4 mg/mL nACP treatment. Data bars denote mean (*n* = 3), and error bars, the standard deviation.

**Figure 6 biomedicines-12-01499-f006:**
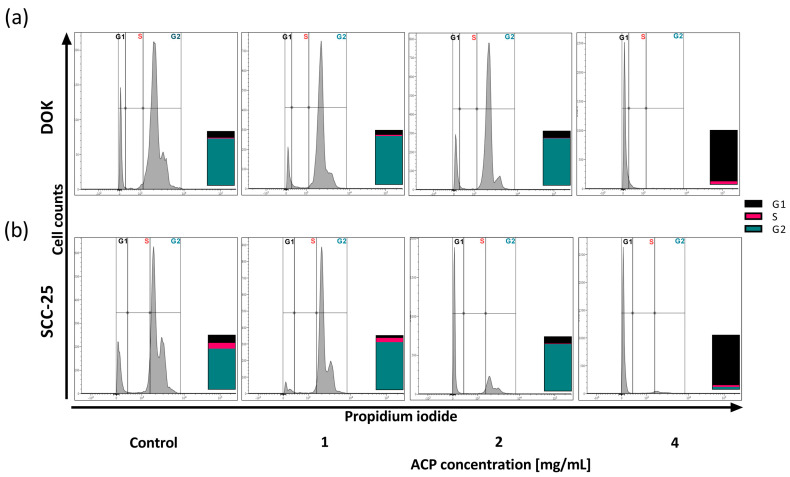
Cell cycle histograms of (**a**) DOK and (**b**) SCC-25 cell lines treated with 1, 2, and 4 mg/mL of nACP for 24 h. Data columns denote mean (*n* = 3).

**Figure 7 biomedicines-12-01499-f007:**
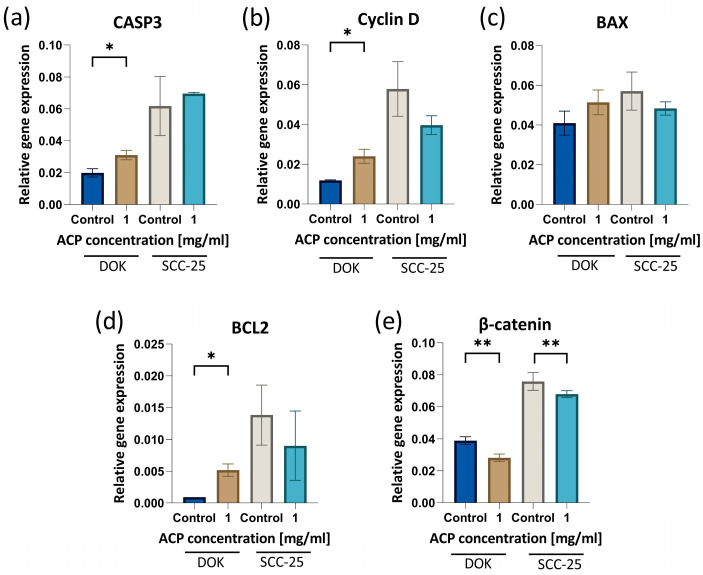
Gene expression analysis of (**a**) CASP3, (**b**) Cyclin D, (**c**) BAX, (**d**) BCL2, and (**e**) β-Catenin of DOK and SCC-25 cell lines treated with 1 mg/mL nACPs for 24 h. * *p*< 0.05, ** *p*< 0.01. Data bars denote mean (*n* = 6), and error bars, the standard deviation.

**Table 1 biomedicines-12-01499-t001:** Primers with corresponding sequences used in the study.

Gene	Forward Primer (5′−3′)	Reverse Primer (5′−3′)
CASP3	TGTTTGTGTGCTTCTGAGCC	CACGCCATGTCATCATCAAC
Cyclin D	CGGAGGAGAACAAACAGATC	GGGTGTGCAAGCCAGGTCCA
BAX	ATGTTTTCTGACGGCAACTTC	AGTCCAATGTCCAGCCCAT
BCL2	ATGTGTGTGGAGAGCGTCAACC	TGAGCAGAGTCTTCAGAGACAGCC
β-Catenin	GCTACTCAAGCTGATTTGATGGA	GGTAGTGGCACCAGAATGGATT
GAPDH	ATGGGGAAGGTGAAGGTCG	GGGGTCATTGATGGCAACAATA

## Data Availability

Data are contained within the article.

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
