# Peer review of "Anti-Cancer Activities of Nano Amorphous Calcium Phosphates toward Premalignant and Oral Cancer Cells"

_biomedicines, 2024, doi:10.3390/biomedicines12071499_

Round 1
Reviewer 1 Report
Comments and Suggestions for Authors
In this article “Anti-cancer activities of nano amorphous calcium phosphates 2 toward premalignant and oral cancer cells” authors addressed the current advancement discovered to treat cancer.. (I) designed ACP nano-sized particles induced limited cytotoxicity against premalignant and malignant OSCC cells, and (II) affected their adhesion. (III) The cellular uptake of nanoparticles was observed.
Authors reported several possible mechanisms of nACP action- including the effect on 107 cell adhesion, invasion, apoptotic and proliferation gene expression, the induction of apoptosis and cell cycle arrest. Therefore, it is a unique theme.
Compared with other published material, it explains Cell adhesion ability after nACP treatment, and proves nanoparticle uptake analysis. The paper is well written. The language is impressive and self-explanatory. The conclusions are consistent with the evidence and arguments presented
I recommend this article for publication. Improve language (English editing required).
Comments on the Quality of English LanguageImprove language (English editing required).
Author Response
Comment 1: In this article “Anti-cancer activities of nano amorphous calcium phosphates 2 toward premalignant and oral cancer cells” authors addressed the current advancement discovered to treat cancer.. (I) designed ACP nano-sized particles induced limited cytotoxicity against premalignant and malignant OSCC cells, and (II) affected their adhesion. (III) The cellular uptake of nanoparticles was observed.
Authors reported several possible mechanisms of nACP action- including the effect on cell adhesion, invasion, apoptotic and proliferation gene expression, the induction of apoptosis and cell cycle arrest. Therefore, it is a unique theme.
Compared with other published material, it explains Cell adhesion ability after nACP treatment, and proves nanoparticle uptake analysis. The paper is well written. The language is impressive and self-explanatory. The conclusions are consistent with the evidence and arguments presented
I recommend this article for publication. Improve language (English editing required).
Answer 1: Thank you for your thorough and positive review of our manuscript. We understand the importance of language precision and have undertaken a comprehensive English editing process to enhance readability and clarity.
Thank you again for your valuable feedback and recommendation.
Reviewer 2 Report
Comments and Suggestions for Authors
Comments to the authors
The authors (Herendija et. al.) have Anti-cancer activities of nano amorphous calcium phosphates 2 toward premalignant and oral cancer cells. However, the manuscript should be revised thoroughly, and further experiments are required to support the hypothesis. Following are some of the comments that the authors might find useful for future submission.
Comment:
- Please describe the method of nanoparticle collection in detail. How will you purify the nanoparticles? Provide more specifics.
- The particle size distribution of ACP showed a relatively uniform distribution (Figure 1c) with a d50 value of 68 nm. The polydispersity index (PDI) should be mentioned.
- Has the author verified the stability of ACP?
- Some figures, such as Figures 4, are unclear and difficult to visualize. Please provide clearer images.
- The characterization of ACP by FTIR is insufficient. NMR data should be provided to confirm the reaction.
- The author should include an FTIR study that compares the modified sample with a control or an unmodified sample.
- The SEM images show a scale of 100 nm, but the particles appear to be much smaller than 10 nm. Please clarify or explain this discrepancy.
- High-resolution SEM or TEM images showing clear morphology of the particles should be provided.
- In the cytotoxicity studies, calculating the IC50 value would be useful. The concentrations of ACP (1, 2, 4 mg/ml) are too high. Please explain.
- No stability studies were conducted. Please include stability studies.
- The reasoning behind the cell cytotoxicity and uptake studies is lacking. The author should provide an explanation with references.
- The manuscript contains grammatical errors and lacks scientific language. Please revise for clarity and accuracy.
Comments on the Quality of English Language
Comments to the authors
The authors (Herendija et. al.) have Anti-cancer activities of nano amorphous calcium phosphates 2 toward premalignant and oral cancer cells. However, the manuscript should be revised thoroughly, and further experiments are required to support the hypothesis. Following are some of the comments that the authors might find useful for future submission.
Comment:
- Please describe the method of nanoparticle collection in detail. How will you purify the nanoparticles? Provide more specifics.
- The particle size distribution of ACP showed a relatively uniform distribution (Figure 1c) with a d50 value of 68 nm. The polydispersity index (PDI) should be mentioned.
- Has the author verified the stability of ACP?
- Some figures, such as Figures 4, are unclear and difficult to visualize. Please provide clearer images.
- The characterization of ACP by FTIR is insufficient. NMR data should be provided to confirm the reaction.
- The author should include an FTIR study that compares the modified sample with a control or an unmodified sample.
- The SEM images show a scale of 100 nm, but the particles appear to be much smaller than 10 nm. Please clarify or explain this discrepancy.
- High-resolution SEM or TEM images showing clear morphology of the particles should be provided.
- In the cytotoxicity studies, calculating the IC50 value would be useful. The concentrations of ACP (1, 2, 4 mg/ml) are too high. Please explain.
- No stability studies were conducted. Please include stability studies.
- The reasoning behind the cell cytotoxicity and uptake studies is lacking. The author should provide an explanation with references.
- The manuscript contains grammatical errors and lacks scientific language. Please revise for clarity and accuracy.
Author Response
The authors (Herendija et. al.) have Anti-cancer activities of nano amorphous calcium phosphates toward premalignant and oral cancer cells. However, the manuscript should be revised thoroughly, and further experiments are required to support the hypothesis. Following are some of the comments that the authors might find useful for future submission.
Comment 1a. Please describe the method of nanoparticle collection in detail. How will you purify the nanoparticles? Provide more specifics.
Answer 1a. In section 2.1 Synthesis of ACP powder, details have been added.
”A previously developed procedure [11] was adapted for the synthesis of ACP powder. ACP was made by abruptly adding a solution containing 23.6 g of Ca(NO3)2 × 4H2O (Sigma Aldrich, Berlin, Germany) in 200 mL of distillated water and 24 mL 28% NH4OH into a solution comprising 6 g of NH4H2PO4, 10 mL 28% NH4OH in 200 mL of distillated water. The fine precipitate formed upon mixing had been aged for 15 s, before it was collected. The sol was centrifuged, washed multiple times until pH 7 was attained. The resulting sol was then poured into Petri (glass) dishes. Finally, the obtained sol was subjected to two-step lyophilization, first at -10 °C and 0.37 mbar for 1 h, and then at -54 °C and 0.1 mbar for 4 h. After lyophilization, the obtained dry powder was transferred to Eppendorf plastic tube (Flex-Tube®).”
Comment 1b. The particle size distribution of ACP showed a relatively uniform distribution (Figure 1c) with a d50 value of 68 nm. The polydispersity index (PDI) should be mentioned.
Answer 1b. d10, d50 and d90 are so-called percentile values. These are statistical parameters that can be read directly from the cumulative particle size distribution. They indicate the size below which 10%, 50% or 90% of all particles are found.
According to the EU recommendation (Commission Recommendation of 10 June 2022 on the definition of nanomaterial (Text with EEA relevance) 2022/C 229/01), the parameter d50 is mainly used as a definition of nano materials (it is necessary that d50 has a value that is less than 100nm).
https://eur-lex.europa.eu/legal-content/EN/TXT/PDF/?uri=CELEX:32022H0614(01)
http://dx.doi.org/10.2760/143118
http://dx.doi.org/10.1016/j.ejpb.2013.03.032
In the section 3.1
Instead of:
”The particle size distribution of ACP determined a relatively uniform distribution (Figure 1c) with a value of d50=68 nm”
Now we inserted:
” Aqueous suspensions of the particles of ACP were analyzed with the aim of establishing their size distributions. Figure 1c shows the size distributions for the suspended particles of ACP powders in distilled water. The particle sizes for the analyzed powder range from 22 nm to 1 μm. The particles had a relative uniform distribution of sizes, with the d50 parameter equaling 68 nm. The powder contained a small portion of larger and smaller particles; correspondingly, d90=147 nm and d10=32 nm.”
Comment 2. Has the author verified the stability of ACP?
Answer 2. Please see the answer to your comment #9.
Comment 3. Some figures, such as Figures 4, are unclear and difficult to visualize. Please provide clearer images.
Answer 3. We have carefully revised Figure 4, and the numbers and letters are now improved for better visualization.
Comment 4. The characterization of ACP by FTIR is insufficient. NMR data should be provided to confirm the reaction.
Answer 4. Thank you for your suggestion. In general, NMR techniques are suitable for the characterization of polymeric biomaterials, while XRD, FTIR, PSD, etc. techniques are preferred when characterizing ceramic biomaterials (https://doi.org/10.1016/j.msec.2017.05.127).
The technique of solid-state NMR is suitable for the characterization of ceramic biomaterials. Unfortunately, solid state NMR is not available to us. In accordance with the aim of the research, we considered that the basic physical and chemical characterization of ACP particles is sufficient (SEM, EDS, PSD and FTIR). The primary objective of this research belongs to biology/medicine (also in accordance with the scope of the journal of Biomedicines) and eight types of biomedical techniques (from 2.2.1 to 2.2.8) were applied for that purpose.
Comment 5. The author should include an FTIR study that compares the modified sample with a control or an unmodified sample.
Answer 5. In our manuscript, we have clearly stated that the primary objective of our study was to investigate whether and how ACP particles affect oral cancer/premalignant cells. This focus necessitates a single-system approach, utilizing only one type of particle: ACP. Consequently, there is no "unmodified sample" in our study, as our research design does not include a control or unmodified comparison group. We hope this clarification helps in understanding that our study is centered exclusively on ACP particles and their interaction with oral cancer cells.
Comment 6. The SEM images show a scale of 100 nm, but the particles appear to be much smaller than 10 nm. Please clarify or explain this discrepancy.
Answer 6. Please see the answer to your comment #1b.
Comment 7. High-resolution SEM or TEM images showing clear morphology of the particles should be provided.
Answer 7. As we mentioned in the answer to your question number 5: In accordance with the aim of the research, we considered that the basic physical and chemical characterization of ACP particles is sufficient (SEM, EDS, PSD and FTIR). The primary objective of this research belongs to biology/medicine (also in accordance with the scope of the journal of Biomedicines) and eight types of biomedical techniques (from 2.2.1 to 2.2.8) were applied for that purpose.
We have already published (in Phys.Chem.Chem.Phys,) a more detailed physicochemical characterization (XRD, SEM, TEM, DTA, TGA) of this system (and systems based on it) in a specialized journal for those purposes (https://doi.org/10.1039/C8CP06460A).
Comment 8. In the cytotoxicity studies, calculating the IC50 value would be useful. The concentrations of ACP (1, 2, 4 mg/ml) are too high. Please explain.
Answer 8. Our aim was to investigate the anti-cancer potential of ACP both as an agent and as a potentially active drug carrier against oral cancer cells, a novel area of research not previously explored in the literature. This study represents the first steps in this field. We aimed to discover and explain the phenomenon and potential of ACP in this context. To guide our experimental design, we adopted the concentrations from the paper available here (https://doi.org/10.3390/cancers15153785) and treated the cancer cells accordingly.
Now that we have established the presence of an anticancer effect, our future studies will focus on the ACP system with varying concentrations, both lower and higher than those used in the current study. The aim of this research was to justify the continuation of our investigations. We will also explore drug-free ACP as well as drug-loaded ACP in our subsequent experiments.
Comment 9. No stability studies were conducted. Please include stability studies.
Answer 9. Thank you for your insightful comment. We would like to clarify that, as described in section 2.1 "Synthesis of ACP Powder," the resulting gel was repeatedly washed with distilled water until the pH of the medium reached 7. This process ensures the attainment of a stable form of ACP, consistent with both literature data and our previous research findings (https://doi.org/10.1007/978-1-4615-5517-9_2, https://doi.org/10.1021/acs.cgd.1c00058, https://doi.org/10.1021/cr0782574, https://doi.org/10.1590/0366-69132021673822965, https://doi.org/10.1039/C8CP06460A, https://doi.org/10.1039/D1TB00601K). We considered that during the experimental period ranging from 1 to 7 days, the freshly synthesized ACP remained stable. In addition, according to the literature, ACP is stable in media with a pH range of 5 to 12 https://doi.org/10.1039/D1BM01239H. All our experiments were conducted in liquid media that ensured the chemical stability of ACP and remained within the specified pH range, which allows for its stability.
Comment 10. The reasoning behind the cell cytotoxicity and uptake studies is lacking. The author should provide an explanation with references.
Answer 10. Our hypothesis was that ACP particles could induce cytotoxic effects on oral cancer and premalignant cells and that these nanoparticles could be efficiently taken up by the cells, thereby influencing their adhesion, proliferation, and viability. The confirmation of this hypothesis through our experiments (MTT, Annexin V, uptake analysis by flow-cytometry) provides a foundation for future research. The cytotoxicity experiments were essential to determine the extent to which ACP could induce cell death in oral cancer cells. The uptake studies, measured via side scatter in flow cytometry, were critical to understanding the interaction between ACP particles and cancer cells. Increased side scatter indicated greater cell complexity, suggesting successful uptake of nanoparticles by the cells, which is an important step in evaluating the potential of ACP as a therapeutic agent. The discussed papers in the manuscript also have cytotoxicity and uptake analyses (References #26,27,28,29,30) and are citated accordingly.
Comment 11. The manuscript contains grammatical errors and lacks scientific language. Please revise for clarity and accuracy.
Answer 11. We acknowledge your concerns regarding the grammatical errors in the manuscript. We have corrected the grammatical errors and ensured that the language is precise and appropriately scientific.
Reviewer 3 Report
Comments and Suggestions for Authors
In section 2.1.1 Powder characterization
The particle size distribution (PSD) is the D50? The size distribution is the span, parameter missing.
In the section 2.2.2. Nanoparticle uptake
non–adherent cells were counted using a hemocytometer. How were the cells collected?
In section 2.2.4. MTT assay
“….was measured at 540 nm using an ELISA reader (RT-2100c, Rayto, 179 China).”
Replace by “….was measured at 540 nm using an microplate reader (RT-2100c, Rayto, 179 China)”
ELISA is a technique not a type of spectrophotometer reader!
Results size distribution with d50=68 nm, please present the span of values obtained by Mastersizer 2000 (Malvern Instruments 128 Ltd., Malvern, UK) to understand the size polydispersivity. Moreover, the nanoparticles from image (Figure (a) FESEM of ACP, sizes are much lower than 68nm!
Figure 2. Cell adhesion assay what is the control of 100% of adhesion?
The cytotoxic effect of ACP nanoparticles on oral premalignant and cancer cells was 278 evaluated 1 day, 3 days, and 7 days after the treatment. The culture medium was replaced each day or each 3 days containing ACP nanoparticles? Please clarify!
In ACP nanoparticle uptake analysis were cells labelled with any fluorochrome? If not how could be measured by flow cytometry?
All figures lack mention of the number of samples analyzed and whether the values presented are the mean and standard deviation.
The value for the different parameters must have the same decimal places e.g. 98%, 71.32%, 32.2%, replace by 98%, 71%, 32%, and also present the standard deviation of the values.
Author Response
Comment 1: In section 2.1.1 Powder characterization. The particle size distribution (PSD) is the D50? The size distribution is the span, parameter missing.
Answer 1: We thank you for the comment. d10, d50 and d90 are so-called percentile values. These are statistical parameters that can be read directly from the cumulative particle size distribution. They indicate the size below which 10%, 50% or 90% of all particles are found.
According to the EU recommendation (Commission Recommendation of 10 June 2022 on the definition of nanomaterial (Text with EEA relevance) 2022/C 229/01), the parameter d50 is mainly used as a definition of nano materials (it is necessary that d50 has a value that is less than 100nm).
https://eur-lex.europa.eu/legal-content/EN/TXT/PDF/?uri=CELEX:32022H0614(01)
http://dx.doi.org/10.2760/143118
http://dx.doi.org/10.1016/j.ejpb.2013.03.032
In the section 3.1
Instead of:
”The particle size distribution of ACP determined a relatively uniform distribution (Figure 1c) with a value of d50=68 nm”
It is now inserted:
”Aqueous suspensions of the particles of ACP were analyzed with the aim of establishing their size distributions. Fig. 1c shows the size distributions for the suspended particles of ACP powders in distilled water. The particle sizes for the analyzed powder range from 22 nm to 1 μm. The particles had a relative uniform distribution of sizes, with the d50 parameter equaling 68 nm. The powder contained a small portion of larger and smaller particles; correspondingly, d90=147 nm and d10=32 nm.”
Comment 2: In the section 2.2.2. Nanoparticle uptake non–adherent cells were counted using a hemocytometer. How were the cells collected?
Answer 2: Non-adherent (dead) cells were floating in the medium, so we gently collected the medium with a pipette. The aspirated cells were then centrifuged and subsequently counted.
Comment 3: In section 2.2.4. MTT assay “….was measured at 540 nm using an ELISA reader (RT-2100c, Rayto, 179 China).” Replace by “….was measured at 540 nm using an microplate reader (RT-2100c, Rayto, 179 China)” ELISA is a technique not a type of spectrophotometer reader!
Answer 3: Thank you for pointing out the inaccuracy. We have corrected the sentence to read: "…was measured at 540 nm using a microplate reader (RT-2100c, Rayto, China)."
Comment 4: Results size distribution with d50=68 nm, please present the span of values obtained by Mastersizer 2000 (Malvern Instruments 128 Ltd., Malvern, UK) to understand the size polydispersivity. Moreover, the nanoparticles from image (Figure (a) FESEM of ACP, sizes are much lower than 68nm!
Answer 4: We completely agree with your comment. We hope we improved the understanding of this result in the previous answer (answer #1 to comment #1).
Comment 5: Figure 2. Cell adhesion assay what is the control of 100% of adhesion?
The cytotoxic effect of ACP nanoparticles on oral premalignant and cancer cells was 278 evaluated 1 day, 3 days, and 7 days after the treatment. The culture medium was replaced each day or each 3 days containing ACP nanoparticles? Please clarify!
Answer 5: The control in cell adhesion assay were the untreated cells (without ACP), the explanation is now included in the Figure 2 legend. As for the MTT assay, the culture medium was replaced each 3 days containing ACP nanoparticles. This clarification is now included in the revised manuscript.
Comment 6: In ACP nanoparticle uptake analysis were cells labelled with any fluorochrome? If not how could be measured by flow cytometry?
Answer 6: In our study, instead of using fluorochrome labeling, we utilized side scatter measurements (which describes the cells complexity) to assess the uptake of ACP nanoparticles by the cells. Indeed, higher side scatter values indicate greater complexity of the cells. In our analysis, we utilized side scatter as a metric to infer the uptake of ACP nanoparticles by cells, as higher side scatter readings suggest increased internalization of the nanoparticles by the cells. This approach provided us with quantitative data on nanoparticle uptake and it is commonly used in similar studies. https://doi.org/10.1002/cyto.a.20927 https://doi.org/10.1039/C8TB01995A https://doi.org/10.1021/acsanm.3c05974
Comment 7: All figures lack mention of the number of samples analyzed and whether the values presented are the mean and standard deviation.
Answer 7: We have revised all figures to include the number of samples analyzed and have specified that the values presented are the mean and standard deviation.
Comment 8: The value for the different parameters must have the same decimal places e.g. 98%, 71.32%, 32.2%, replace by 98%, 71%, 32%, and also present the standard deviation of the values.
Answer 8: We have revised the manuscript to ensure that the values for the different parameters have consistent decimal places. Additionally, we have included the standard deviations for these values to provide a clearer representation of the data.
Round 2
Reviewer 3 Report
Comments and Suggestions for Authors
Although the authors did not consider all suggestions for improving the manuscript, it can be considered for publication.